# Comparing ecosystem gaseous elemental mercury fluxes over a deciduous and coniferous forest

Jun Zhou [1,2], Silas W. Bollen[1], Eric M. Roy [1,3], David Y. Hollinger[4], Ting Wang[1], John T. Lee[5] & Daniel Obrist [1,6] ✉

Sources of neurotoxic mercury in forests are dominated by atmospheric gaseous elemental mercury (GEM) deposition, but a dearth of direct GEM exchange measurements causes major uncertainties about processes that determine GEM sinks. Here we present three years of forest-level GEM deposition measurements in a coniferous forest and a deciduous forest in northeastern USA, along with flux partitioning into canopy and forest floor contributions. Annual GEM deposition is $13.4 \pm 0.80$ µg m$^{-2}$ (coniferous forest) and $25.1 \pm 2.4$ µg m$^{-2}$ (deciduous forest) dominating mercury inputs (62 and 76% of total deposition). GEM uptake dominates in daytime during active vegetation periods and correlates with $CO_2$ assimilation, attributable to plant stomatal uptake of mercury. Non-stomatal GEM deposition occurs in the coniferous canopy during nights and to the forest floor in the deciduous forest and accounts for 24 and 39% of GEM deposition, respectively. Our study shows that GEM deposition includes various pathways and is highly ecosystem-specific, which complicates global constraints of terrestrial GEM sinks.

Mercury (Hg) is of widespread environmental concern because it undergoes long-range air transport and after deposition, bioaccumulates in food chains to exert neurotoxic, cardiovascular, and reproductive harm[1,2]. In terrestrial ecosystems, dry deposition of atmospheric gaseous elemental Hg (GEM) is considered the dominant Hg source, driven largely by vegetation uptake and subsequent transfer to soils by litterfall[3,4]. Terrestrial GEM deposition results in accumulation of Hg in soils[5], and thereafter can bioaccumulate in terrestrial biota[6,7] and propagate through watersheds[4].

Many studies have investigated the atmospheric Hg deposition and soil emission fluxes and contributed significantly to the understanding of Hg dynamics in the forests and the role of forests in global mercury cycling since 2000[8-12]. Recent datasets showed the global vegetation GEM uptake is estimated at $2{,}705 \pm 504$ Mg yr$^{-1}$ and mainly

driven by forests[3,4,13]. While vegetation GEM uptake has been quantified in different plant functional groups and across various ecosystems[14], it is not equal to whole ecosystem GEM loads which complicates estimation of terrestrial Hg deposition mass balances. A reason is that vegetation uptake may not account for GEM flux contributions from non-active vegetation periods, non-photosynthetically active tissues (i.e., woody tissues), and forest floors (soils and litter). For example, while at the ecosystem-level, two rural forests (e.g., in Connecticut and Massachusetts, USA) showed largely GEM sinks linked to vegetation uptake, substantial GEM emission[15] were reported from mildly and moderately polluted subtropical forests driven by re-emissions from soils or plant surfaces back to the atmosphere. Because these forest studies did not specifically quantify GEM exchange over forest floors[16], which can serve either as sinks and sources[17-19], we

[1]Department of Environmental, Earth and Atmospheric Sciences, University of Massachusetts, Lowell, MA, USA. [2]Key Laboratory of Soil Environment and Pollution Remediation, Institute of Soil Science, Chinese Academy of Sciences, Nanjing 210008, China. [3]Department of Earth, Atmospheric and Planetary Sciences, Massachusetts Institute of Technology, Cambridge, MA, USA. [4]USDA Forest Service, Northern Research Station, Durham, NH, USA. [5]School of Forest Resources, University of Maine, Orono, ME, USA. [6]University of California, Agriculture and Natural Resources, Davis, CA, USA. ✉e-mail: daniel_obrist@uml.edu

currently lack a comprehensive understanding of how forest type and structure affect ecosystem-level forest GEM exchanges and how they partition into canopy and forest floor contributions seasonally and diurnally.

The scientific objectives of this study were to investigate forest-level GEM exchanges and their partitioning into canopy and forest floor contributions in both a deciduous and coniferous rural forest in northeastern USA, and assess underlying mechanisms and drivers of GEM exchange in the two different forest types by analyzing seasonal, daytime, and nighttime contributions and relationships to $CO_2$ assimilation. Spatial and temporal partitioning of fluxes allows us to infer the impacts of stomatal uptake (e.g. during daytime in canopies) and non-stomatal contributions (e.g., nighttime and forest floor exchange) to ecosystem GEM deposition. We compare a 470-day record of ecosystem-level GEM exchange of a mid-latitude deciduous forest in Massachusetts (May 2019 to August 2020)[3] with a nearby (410 km distance) 560-day coniferous forest record in Maine, USA (November 2020 to May 2022). The two records are the first to directly compare GEM deposition patterns of two different forest types and complement only a handful of other studies that report annual GEM deposition patterns (e.g., grasslands[20–22], tundra[23], agricultural field[24], northern peatland[11]). All fluxes were measured using tower-based micrometeorological flux-gradient approaches, whereby GEM gradient measurements were deployed both above the forest canopies and additionally above forest floors. The set-up allowed quantification of whole-ecosystem (above canopy) and forest floor GEM fluxes, and by difference canopy contributions to GEM exchange, when combined with quantification of correspondingly measured atmospheric turbulence parameters.

## Results and discussions
### Whole-ecosystem forest GEM exchange and correlation with $CO_2$ assimilation

Both forests exhibited inherent similarities of annual and seasonal GEM exchange patterns, with GEM deposition dominating overall

resulting in net annual GEM deposition in both forests. Annual GEM uptake in both forests was dominated by deposition during active vegetation periods, while GEM emissions generally occurred during winter and spring periods (Fig. 1 and S1). As evidence of an active role of vegetation GEM uptake, GEM fluxes were positively correlated with forest leaf area indices at both forests during active vegetation periods (LAI; $r^2$ of 0.61 in the deciduous forest not including November; $r^2$ of 0.51 in the coniferous forest not including March and April).

Notable differences in the GEM flux records were observed between the forests, although the two observational sets were performed during subsequent years so that reasons for differences may include inter-annual variability in flux behavior. The first notable difference is that growing-season GEM deposition in the coniferous forest started earlier (March 1st and March 3rd in year 1 and 2, respectively) than in the deciduous forest (June 26th and June 1st, respectively) (Figs. 1, 2, and S2). We similarly observed an earlier onset of daily $CO_2$ uptake between the two forests, whereby consistent seasonal $CO_2$ uptake starting on April 3rd in the coniferous forest and on June 2nd and 5th in the deciduous forest[25,26]. In this deciduous forest, we previously reported growing-season correlation of GEM exchange with ecosystem $CO_2$ exchange[3], which we attributed to foliar GEM uptake[3,17,28] and is consistent with studies showing that Hg in foliage is largely derived from atmospheric GEM uptake[4,29–31]. In the coniferous forest, daily GEM sinks preceded those of $CO_2$ by about one month, which may be explained by an earlier onset of plant photosynthesis in comparison to net daily $CO_2$ uptake, which is delayed in time due to nighttime respiration.

A second difference between the two forests is the GEM deposition magnitude with 30-min. fluxes that showed significantly higher values in the deciduous forest compared to those in the coniferous forest ($p < 0.01$). Annual GEM deposition in the deciduous forest of $25.1\,\mu g\,m^{-2}$ (95% CI: $23.2$–$26.7\,\mu g\,m^{-2}$) was almost twice the GEM deposition in the coniferous forest of $13.4\,\mu g\,m^{-2}$ (95% CI: $12.0$–$14.3\,\mu g\,m^{-2}$). One reason is that the underlying forest floor showed a deposition in the deciduous forest and an emission in the coniferous

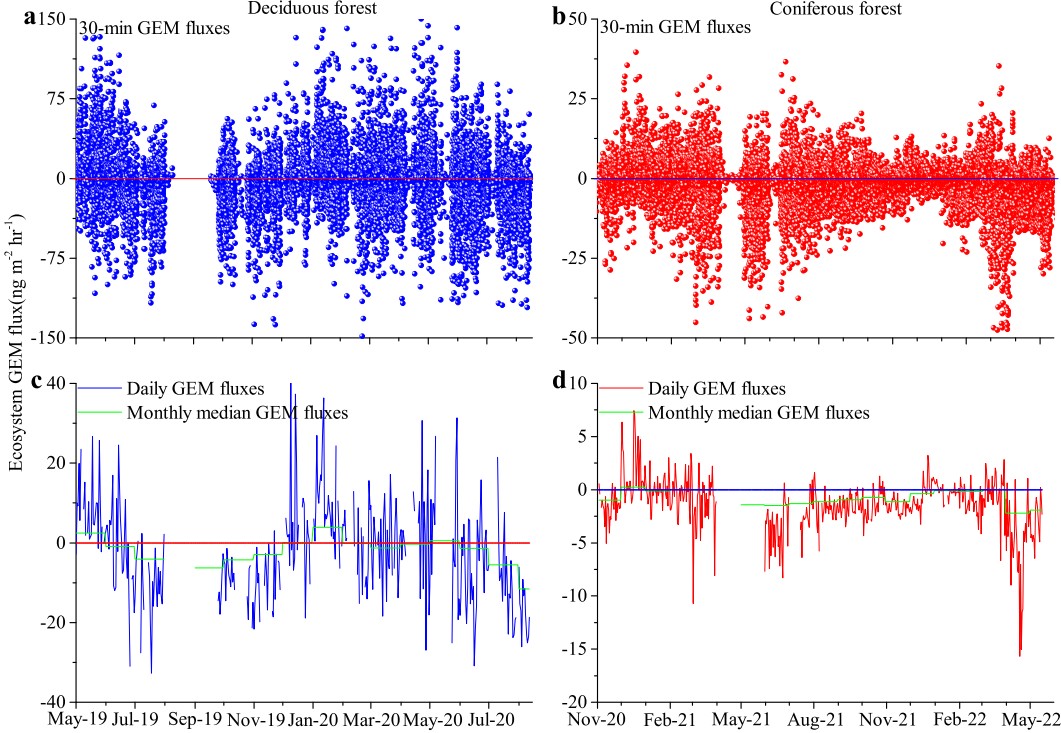

**Fig. 1 | Whole ecosystem gaseous elemental mercury (GEM) fluxes.** Thirty-minute resolution GEM exchange fluxes (**a**, **b**), daily mean GEM fluxes and median monthly GEM fluxes (**c**, **d**) measured in the deciduous forest (**a** and **c**) and coniferous forest (**b** and **d**). Negative GEM fluxes denote deposition and positive fluxes represent emissions.

 

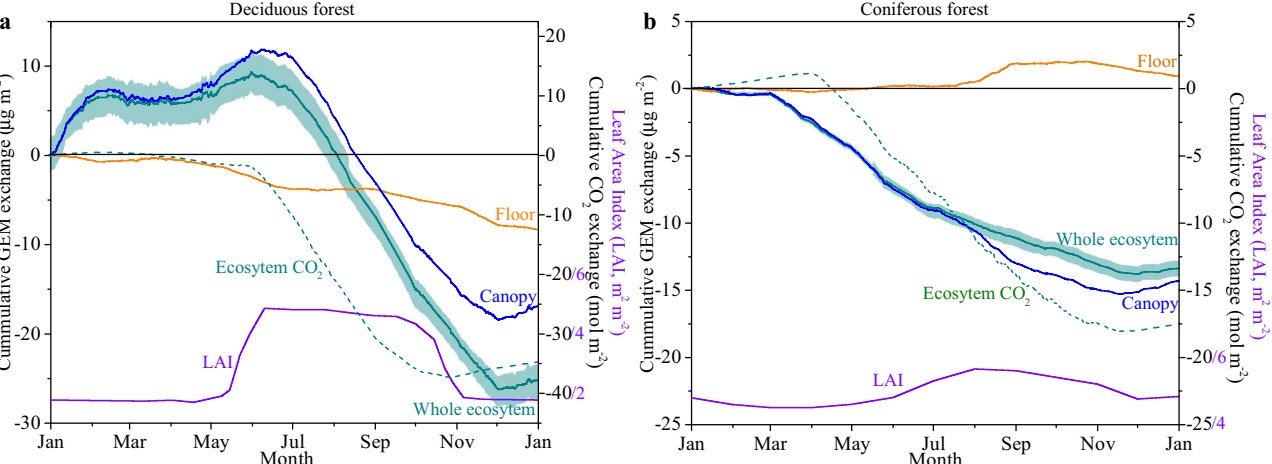

**Fig. 2 | Cumulative annual gaseous elemental mercury (GEM) flux patterns.**
Cumulative annual gaseous elemental mercury (GEM) flux patterns measured at the whole ecosystem (green lines) and above forest floors (orange lines), calculated for forest canopies (blue lines), leaf area index (LAI), and $CO_2$ fluxes for the deciduous forest, which correspondingly increased or decreased ecosystem-level deposition (discussed in the next section). A higher net ecosystem productivity (NEP, i.e., cumulative net $CO_2$ deposition) likely is another reason for higher GEM deposition in the deciduous forest. Indeed, the difference in GEM flux was similar to the difference in annual NEP (factor 2.1), which amounted to 416 g C m$^{-2}$ yr$^{-1}$ in the deciduous forest and 197 g C m$^{-2}$ yr$^{-1}$ in the coniferous forest[25,32]. This suggested that ecosystem NEP may be a proxy for the magnitude of annual GEM deposition, a notion which we explored further by combining our two forest flux records with three additional ecosystems flux records that report annual GEM and NEP magnitudes (including an evergreen broadleaf forest, a grassland, and a tundra flux record; Table 1). Our analysis shows that the ratios of annual ecosystem GEM to NEP exchanges are highly variable and even inconsistent in directions among the sites (ranging from −0.073 to +0.089 µg Hg g$^{-1}$ C). A reason for this variability is that GEM and NEP fluxes only correlate during growing seasons (June-October in the deciduous forest and April-October in the coniferous forest), while wintertime periods show a decoupling of the two gaseous exchange fluxes. For example, wintertime NEP patterns consistently show respiration-driven emissions of carbon across all sites. GEM fluxes, however, show site-specific wintertime patterns that range from emissions (e.g., in the deciduous forest and the grassland) to deposition (e.g., in the coniferous forest and the arctic tundra). When considering growing season data only, cumulative GEM deposition and NEP significantly and linearly correlate in our two forest sites ($r^2 = 0.95$ for both forests) showing ecosystem Hg:C uptake ratios of 0.056 µg g$^{-1}$ (coniferous forest) and 0.091 µg g$^{-1}$ (deciduous forest). Other sites showed growing season GEM and C uptake ratios of 0.021 µg g$^{-1}$ (grassland), 0.033 µg g$^{-1}$ (tundra), and 0.067 (subtropical broadleaf forest) (Table 1). While these ratios are highly variable, the order of GEM:C uptake ratios is linearly correlated with GEM uptake rates in leaves of dominant species in these ecosystems ($r^2$ of 0.92, $p < 0.01$) (Table 1). The ratios are also consistent with generally lower Hg concentrations observed in grassland species compared to tree foliage[33] and with higher Hg uptake capacity in deciduous leaves compared to coniferous needles[34].

Atmosphere-surface GEM exchange fluxes measured by the gradient method at 30-min. resolution are highly variable and frequently alternate between emission (positive values) and deposition (negative values) in both forests (Fig. 1). Reasons for this variability

and coniferous (**b**) forests. Double measured months in subsequent years were averaged to display annual sums and the original 15- and 18-months cumulative measurement records are shown in Fig. S4.

are that measurement of small GEM fluxes against the relatively larger atmospheric background is challenging at the level of background ecosystems and results in substantial flux variability and uncertainty ranges (Text S1 and Fig. 1A, B). Repeatable seasonal and diel patterns (Fig. 1 and S2) and error propagation based on random error analysis and standard deviation of the uncertainty of time-repeated measurements (Text S3, Fig. S2), however, provide confidence in the reliability of our GEM flux measurements. Monthly and cumulative GEM exchange fluxes also show similar seasonal flux patterns between the 2 years of measurements in both forests, albeit with some monthly differences (Fig. S3). For example, in the deciduous forest, GEM showed emission in May of the first year (7.2 ng m$^{-2}$ hr$^{-1}$) yet small deposition in the second year (−0.95 ng m$^{-2}$ hr$^{-1}$), and a much larger deposition occurred in March of the second year (−4.5 ng m$^{-2}$ hr$^{-1}$) compared to the first year (−1.1 ng m$^{-2}$ hr$^{-1}$). These differences suggest a presence of inter-annual variability in forest-level GEM fluxes in response to potential difference in environmental conditions, plant physiology, and ecosystem properties, although longer-term measurements are needed to quantify inter-annual variability in GEM exchanges across ecosystems.

## Flux partitioning into canopy and forest floor contributions

Two additional factors likely weaken relationships of ecosystem $CO_2$ fluxes and GEM deposition, including forest floor GEM flux contributions (soils and litter; discussed here) and nighttime GEM exchanges (discussed in section 3.3). At both forests, we deployed a second flux-gradient system under canopies to quantify forest-floor GEM exchanges and partition ecosystem-level exchanges into forest floor and canopy contributions (difference between whole-ecosystem and floor GEM exchanges). Our data show that the forest floor (combined soil, litter, and ground moss) was a consistent GEM sink (i.e., deposition) in the deciduous forest accounting for annual GEM deposition of 9.7 µg m$^{-2}$. In contrast, the coniferous forest floor acted as a small GEM source of 0.93 µg m$^{-2}$ (Fig. 2 and S4), and these differences in forest floor behaviors explain a substantial part of observed difference in annual ecosystem GEM deposition. The differentials between forest floor and whole ecosystem fluxes represent canopy contributions, which show surprisingly similar annual GEM deposition of 14.3 µg m$^{-2}$ in the coniferous forest and 15.4 µg m$^{-2}$ in the deciduous forest (only a 7.7% difference). Forest floor fluxes account for 6.5% (opposite direction) and 63% (same

**Table 1 | Summary of annual and growing season gaseous elemental mercury (GEM) deposition (µg m$^{-2}$), CO$_2$ uptake (i.e., net ecosystem productivity (NEP)) (g C m$^{-2}$), GEM and NEP uptake ratios (µg g$^{-1}$), and foliar Hg uptake rates of dominant plants across five ecosystems**

| Biomes | Growing season GEM deposition | Growing season NEP | Growing season ratio of GEM deposition:NEP | Annual GEM deposition | Annual NEP | Annual GEM deposition:NEP | Foliar Hg:C concentration ratios | Foliar Hg accumulation rate (ng Hg g$^{-1}$ mon$^{-1}$) | References |
|---|---|---|---|---|---|---|---|---|---|
| Evergreen broadleaf | 53.9 | 800 | 0.067 | 53.9 | 800 | 0.067 | 0.108 | 4.1 | Wang et al.[56] |
| Deciduous | 35.3 | 390 | 0.091 | 34.9 | 416 | 0.084 | 0.082 | 5.1 | This study |
| Coniferous | 12.6 | 224 | 0.056 | 13.4 | 210 | 0.064 | 0.067 | 3.1 | This study |
| Grass | 37.67 | 1778.6 | 0.021 | 25.1 | 946 | 0.027 | 0.040 | 2.5 | Fritsche et al.[22] |
| Tundra | 3.56 | 108.5 | 0.033 | 6.5 | -89 | -0.073 | 0.067 | 2.9 | Obrist et al.[23] |

direction) of canopy-level GEM fluxes in the coniferous and deciduous forests, respectively, showing variable magnitudes and directions of forest floor contributions to ecosystem-level GEM exchanges. Hence, characterizing forest floor GEM exchanges will be critical measures to constrain whole-ecosystem Hg budgets across ecosystems. Calculated canopy GEM fluxes (i.e., by difference) exhibited diurnal variability, and as expected are more strongly correlated with CO$_2$ fluxes during growing seasons compared to whole ecosystem fluxes (Fig. 3). Outside of active growing seasons, canopy fluxes showed GEM emission to the atmosphere, which was enhanced during daytime and is in support of photochemically driven GEM emission processes, similar to such emissions reported from many soils. Overall, the patterns of GEM fluxes at these forests are consistent with a highly physiologically (i.e., stomatally) controlled leaf GEM uptake processes as proposed in previous studies[34,35]. Estimated cumulative canopy-only GEM fluxes show some emission processes during parts of the year in both forests which is consistent with the notion that foliage GEM exchange is a bidirectional exchange process[36], and therefore supports a recent isotopic study that indicated 30% of leaf Hg taken up may not permanently and structurally be integrated in leaves but may re-emit back to the air by reduction of Hg$^{2+}$ bound in the leaf interior[17].

**Flux partitioning into daytime and nighttime contributions**

We partitioned GEM exchanges into day- and night-time contributions using photosynthetically active radiation (PAR) measurements. In the deciduous forest, we observed that daytime GEM uptake was dominant, with annual cumulative daytime GEM sinks of 22.6 µg m$^{-2}$ (whole ecosystem) and 21.3 µg m$^{-2}$ (canopy only) which accounted for 90 and 143% of total ecosystem and canopy deposition, respectively (Fig. S5). During nighttime, the deciduous ecosystem showed a cumulative net deposition of 2.5 µg m$^{-2}$ but the canopy showed a cumulative net emission of 3.2 µg m$^{-2}$ that accounts for about 30% of the magnitude of observed daytime deposition (albeit in different direction). The coniferous forest showed distinctly different patterns, whereby daytime GEM uptake accounted for only 73 and 76% of total ecosystem- and canopy-level GEM uptake. Here, nighttime GEM exchange exhibited an additional GEM deposition of 2.9 µg m$^{-2}$ (whole-ecosystem) and 3.0 µg m$^{-2}$ (canopy), indicating an important contribution of non-stomatal GEM uptake in coniferous foliage accounting for about a quarter of the coniferous forest GEM sink. Non-stomatal uptake pathways have been suggested previously based on laboratory studies[37–40] and vertical GEM gradients in forest canopies along with stable isotope signatures[41], but to our knowledge have not been directly measured in situ in ecosystems before.

The nighttime GEM sinks measured in the coniferous forest is independently supported by observed atmospheric GEM concentration declines in the lower atmosphere during nights. Ecosystem GEM uptake in combination with stable nocturnal boundary layer conditions shows substantial GEM depletions in near-surface atmospheric GEM concentrations (Fig. S6). The observation of substantial nighttime GEM deposition in the coniferous forest may lead to higher observed throughfall Hg concentration and throughfall Hg deposition in coniferous forests compared to deciduous forests[5], and would be consistent with a recent stable Hg isotope study which suggest that 34 – 82% of Hg in throughfall might originate from atmospheric GEM uptake[27]. In our study, we measured high throughfall Hg concentrations in the coniferous forest of 28.8 ± 17.0 ng L$^{-1}$ and estimate annual throughfall Hg deposition of 8.5 µg m$^{-2}$ yr$^{-1}$, which strongly exceeds open-field precipitation Hg concentration (5.3 ± 2.1 ng L$^{-1}$) and open-field wet deposition (5.1 µg m$^{-2}$ yr$^{-1}$). Hence, we propose that a substantial part of measured ecosystem GEM deposition (about 16%) maybe be subject to throughfall deposition, as opposed to depositing to ecosystems only via litterfall.

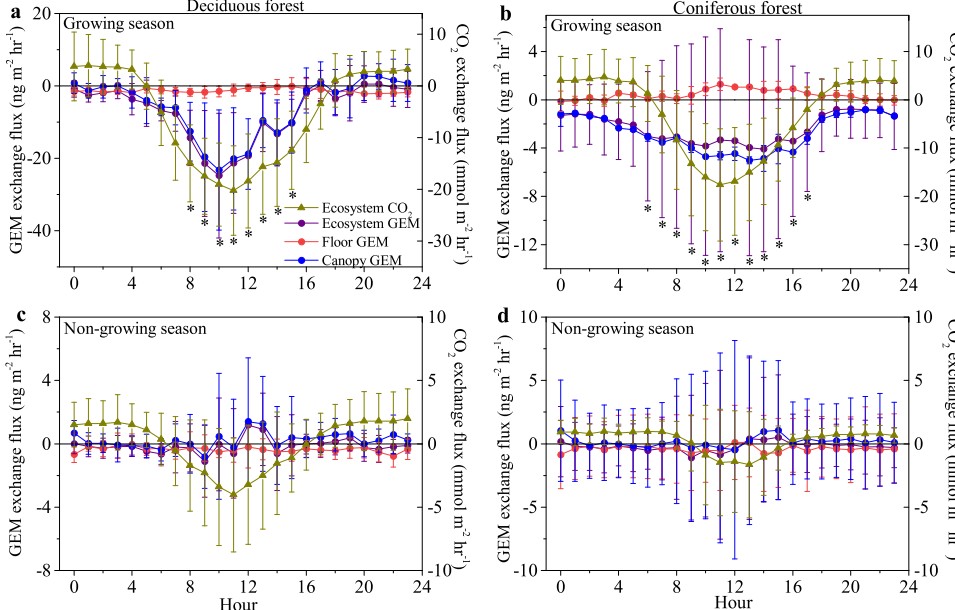

**Fig. 3 | Diel patterns of gaseous elemental mercury (GEM) fluxes.** Diel patterns of gaseous elemental mercury (GEM) fluxes for ecosystem-level, canopy, and forest floors, along with corresponding ecosystem $CO_2$ fluxes for the active growing season and non-growing season in the deciduous forest (**a**, **c**) and coniferous forest

(**b**, **d**). Bars represent standard deviation of hourly replicates ($n > 90$). * represent statistically significant ecosystem fluxes for each hour at the $p < 0.05$ level. The differences of the hourly flux data between the two forests during the growing season are shown in Fig. S11.

## Implications for forest Hg mass balance

Mass balance estimates suggest that GEM dry deposition was the dominant Hg deposition pathway in both the coniferous and deciduous forest, accounting for 62% (coniferous forest) and 76% (deciduous forests) of total atmospheric Hg deposition (Table 2). Hg wet deposition by rain and snow amount to $5.1\,\mu g\,m^{-2}\,yr^{-1}$ at Howland forest, in similar range to that at the Harvard forest ($5.0\,\mu g\,m^{-2}\,yr^{-1}$)

**Table 2 | Summary of estimated and measured gaseous elemental mercury (GEM), particulate mercury (PHg), reactive gaseous mercury (Hg(II)) deposition processes (negative values), mercury emissions and re-emissions to the atmosphere (positive values) in the coniferous and deciduous forests of our study**

| Category | Deciduous forest (Harvard Forest) | | Coniferous forest (Howland Forest) | |
|---|---|---|---|---|
| | Concentration | Annual fluxes ($\mu g\,m^{-2}$) | Concentration | Annual fluxes ($\mu g\,m^{-2}$) |
| Annual gaseous elemental mercury (GEM) | 1.10 ng m$^{-3}$ | −25.1 | 1.03 ng m$^{-3}$ | −13.4 |
| Growing season GEM | 1.10 ng m$^{-3}$ | −21.9 | 0.99 ng m$^{-3}$ | −12.6 |
| Canopy GEM | 1.10 ng m$^{-3}$ | −15.4 | 0.99 ng m$^{-3}$ | −14.3 |
| Forest floor GEM | | −9.7 | | 0.93 |
| Litterfall | 41.1 ng g$^{-1}$ | −12.3$^b$ | 33.4 ng g$^{-1}$ | −11.2 |
| Precipitation | | ~−5 (NADP) | 4.3 ng L$^{-1}$ | −5.1 |
| Throughfall | | −7.0$^2$ | 7.5 ng L$^{-1}$ | −8.8 |
| Hg(II)$^a$ | 4.1 pg m$^{-3}$ | ~−1.9 | | ~−1.9 |
| PHg$^a$ | | ~−1.1 | | ~−1.1 |
| Total flux | | −33.1 | | −21.5 |
| % GEM of total deposition | | 76% | | 62% |

$^a$ Estimated from Obrist et al.[3].
$^b$ Estimated from Risch et al.[45].

estimated by interpolated maps from the National Atmospheric Deposition Program[42]. We estimate Hg(II) dry deposition of $1.9\,\mu g\,m^{-2}\,yr^{-1}$ at both forests based on direct measurements of Hg(II) concentrations at the Harvard Forest[3] and similar geographic locations. We assume $1.1\,\mu g\,m^{-2}\,yr^{-1}$ of particulate Hg (PHg) dry deposition based on average deposition rates reported across North America. Hence, total Hg deposition is estimated at $33.1\,\mu g\,m^{-2}\,yr^{-1}$ and $21.5\,\mu g\,m^{-2}\,yr^{-1}$ in the two forests, of which total dry deposition is estimated at 28.1 and $16.4\,\mu g\,m^{-2}\,yr^{-1}$, respectively. The estimated GEM dry deposition and total dry deposition in the coniferous forest were similar to the modeling estimation across North America with the ranges of $15\text{-}22\,\mu g\,m^{-2}\,yr^{-1}$ over coniferous forests, but much higher than the modeling estimation with the mean ranges of $2\text{-}15\,\mu g\,m^{-2}\,yr^{-1}$ over deciduous forest[43]. Going forward, model parameterization and whole ecosystem GEM flux measurements need to be reconciled.

In the absence of direct GEM flux measurements, dry deposition can be estimated based on litterfall and throughfall deposition (dry deposition = throughfall + litterfall loadings − open area loadings)[44]. Litterfall Hg deposition is estimated at $11.2\,\mu g\,m^{-2}\,yr^{-1}$ at Howland Forest based on foliar concentrations measurements at the end of the growing season and using 10-year average litterfall mass from the site, and estimated to be similar ($12.3\,\mu g\,m^{-2}\,yr^{-1}$) at Harvard forest based on average litterfall Hg deposition estimates for deciduous forests across eastern USA[45]. Based on repeated throughfall collections, we measured Hg throughfall concentrations averaging $7.5\,ng\,L^{-1}$ which yields an estimated throughfall deposition flux of $8.8\,\mu g\,m^{-2}\,yr^{-1}$ when extrapolating to a full year in the coniferous forest. Similarly, throughfall Hg deposition fluxes of $7.0\,\mu g\,m^{-2}\,yr^{-1}$ were estimated in deciduous forests using widespread throughfall Hg deposition measurements across the eastern USA[45]. Using these proxy measurements, we estimate total Hg dry deposition of $14.3\,\mu g\,m^{-2}\,yr^{-1}$ (deciduous forest) and $14.9\,\mu g\,m^{-2}\,yr^{-1}$ (coniferous forest) in the two forests, in comparison to $33.1\,\mu g\,m^{-2}\,yr^{-1}$ and $21.5\,\mu g\,m^{-2}\,yr^{-1}$ when measured directly including measurement of GEM depositions. For the coniferous forest, this estimate is indeed close to estimated dry deposition with direct GEM flux measurements ($16.4\,\mu g\,m^{-2}\,yr^{-1}$). For the deciduous forest, the proxy dry deposition estimate is much lower than

measurements that include direct GEM flux quantification. Reasons for the discrepancies include a substantial GEM deposition to forest floors in the deciduous forest (9.7 µg m$^{-2}$ yr$^{-1}$). Proxy Hg dry deposition estimates, however, closely match estimated canopy GEM fluxes based on our flux partitioning in both forests. We propose that combined litterfall and throughfall deposition hence may be a good estimate for total canopy deposition, yet may underestimate ecosystem deposition when forest floors serve as additional GEM sinks.

In summary, our results show that ecosystem-level GEM deposition in forests includes complex patterns of multiple deposition pathways and GEM sinks located in different ecosystem compartments varying over time. The complexity of deposition pathways represents a challenge in building mass balance of Hg deposition in forests and in scaling up GEM sinks across global forests, which are considered the dominant GEM sinks globally.

## Methods

### Study site

The two research sites are a coniferous forest in Maine, USA and a deciduous forest in Massachusetts, USA. In the coniferous forest, GEM fluxes were measured continuously from November 3, 2020 to May 15, 2022 (18 months) at the Howland Forest AmeriFlux site. The site is 56 km north of Bangor, Maine, USA (45.2 °N 68.7° W) at an elevation 60 m, with a mean annual temperature of 6.1 °C and annual average precipitation of 1148 mm[46]. The forest stand is dominated by red spruce (*Picea rubens* Sarg.) and eastern hemlock (*Tsuga canadensis* (L.) Carr.) which account for 41 and 25% of basal area, with other coniferous species (e.g., northern white cedar, *Thuja occidentalis*; white pine, *Pinus strobus*; balsam fir *Abies balsamea*) and hardwoods (e.g., red maple, *Acer rubrum*; paper birch, *Betula papyrifera*) accounting for 23 and 11% of basal area, respectively[47]. Average canopy height is 23 m with a mean stand age of 120 years and maximum of 225 years. Leaf area index (LAI) of the evergreen canopy showed little seasonal variations and peaked during the growing season with an LAI of 5.8 ± 0.5 m$^2$ m$^{-2}$ (ref. 25). Soils are spodosols formed in well to poorly drained glacial till and high acidity due to coniferous litterfall inputs. GEM flux measurements were conducted on a 32-m tall tower near a climate-controlled shed that housed instrumentation. Near-continuous forest extends several kilometers in all directions of the tower with no occupied dwelling within 6 km distance, and the area beyond is sparsely populated. The closest highway is about 4 km distant. More information on forest stand, foliage dynamics and soil structure are found in recent publications[25,48].

The comparison data set from the deciduous forest was measured at Harvard Forest in Massachusetts, USA (42.5° N, 72.2° W) and was published previously[3]. In short, the forest extends across 1,500 ha with a canopy height of about 24 m. The forest type is dominated by red oak (*Quercus rubra*) along with red maple (*Acer rubrum*), white pine (*Pinus strobus*), and eastern hemlock (*Tsuga canadensis*). Nearly all forests in the region are second-growth forests after large-scale forest clearing in the mid-1800s. GEM flux measurement was conducted on the 31-m-tall flux tower near a climate-controlled hut from May 1, 2019 to August 12, 2020.

### Micrometeorological GEM flux measurements

The sites at Harvard and Howland forests are home to the two longest micrometeorological Eddy Covariance (EC) records of $CO_2$ in the world, extending back to 1990 and 1996, respectively. Micrometeorological techniques for GEM flux measurements also include EC and a similar method, the relaxed eddy accumulation (REA) method, which both require high-resolution controls of GEM measurements[49]. Other methods include aerodynamic gradient (AGM) and modified Bowen ratios (MBR) methods that quantify concentration gradients of trace gases, with each method exhibiting both benefits and drawbacks as reviewed by Sommar et al.[10,50].

GEM flux measurements were conducted at the two forests for a total of nearly 34 months using the AGM method (or flux-gradient method). At Harvard Forest, the approach is described in detail in Obrist et al.[3], and a similar set-up was used at Howland forest. Briefly, GEM fluxes are based on measured GEM gradients at two vertical heights above the forest canopy on a large tower and multiplied by a measure of atmospheric turbulence (i.e., eddy diffusivity K) as described by Edwards et al.[51]. The calculation of GEM flux follows equation:

$$F_{GEM} = -K \frac{\Delta C_{GEM}}{\Delta z} = \frac{u^* k (C_2 - C_1)}{\ln(\frac{z_2 - d}{z_1 - d}) - \Psi_{h2} + \Psi_{h1}} \qquad (1)$$

where $k$ is the von Karman constant (0.4); $u^*$ is the friction velocity; $C_2$ and $C_1$ are the concentrations measured at the upper (height $z_2$) and lower (height $z_1$) inlets, respectively; $d$ is the zero-plane displacement height; and $\Psi_{h1}$ and $\Psi_{h2}$ are the integrated similarity functions for heat at $z_1$ and $z_2$. The $\Psi_h$ is stability dependent and derived as shown in previous studies[51,52]. The total integral footprint that contributes to the concentration profile formation largely originates within a 500-m radius of the measurement tower[53].

At Howland Forest, an EC system measured continuous concentrations of $CO_2$, $H_2O$, and $CH_4$, at the top of the tower with a cavity ring-down spectrometer (model G2311-f, Picarro Inc., Santa Clara, CA, USA). Sensible heat flux and other micrometeorological data needed to derive $K$ and $\Psi$ parameters in Eq. 1 were measured with a SAT-211/3 K 3-axis sonic anemometer (Applied Technologies Inc., Longmont, CO). Under the forest canopy, a second GEM flux system was installed on a small tower (about 3 m) above the forest floor consisting of an additional sonic anemometer (Model 81000, R. M. Young Company, Traverse City, Michigan) at height of 2.6 m to measure the micrometeorological parameters that were used to calculate the $K$ and $\Psi$ parameters. Here, we also measured GEM vertical gradients and flux calculations were performed similar to those above the forest canopy (Eq. 1). Similar instrumentation and measurement set-ups were used at Harvard forest as described in Obrist et al.[3].

GEM concentration differences ($C_2$-$C_1$ in Eq. 1) were measured using two Mercury Vapor Analyzers (Model 2537X and 2537B, Tekran Inc., Toronto, Canada), one for above-canopy (whole-ecosystem) fluxes and a second for forest-floor GEM fluxes. Synchronized two port sampling systems (Model 1110, Tekran Inc.) were used to switch between the two gradient inlets every 10 min. Ecosystem GEM gradients were measured at heights of 28.2 m and 23.2 m, and forest-floor GEM gradients were measured at 1.3 m and 2.6 m above the floor at the Howland Forest. At Harvard Forest, the measurement heights were 24.1 m and 30.8 m (and 29 m before May 15, 2019) for ecosystem GEM gradients and 0.4 m and 1.2 m for forest-floor GEM gradients. Lines not sampled were flushed by a pump to avoid stagnant air. Closed path infrared gas analyzers (LI 7810, LiCor Inc., Lincoln, NB) were used for corresponding measurements of $CO_2$ concentration gradients and used to validate gradient-flux calculations of $CO_2$ by comparison of gradient-flux results with corresponding EC fluxes. Service visits were performed every 3–5 weeks at Howland forest to calibrate systems, perform leak and contamination tests, and rotate inlet lines to prevent null-gradients, similar to work described for the Harvard Forest site[3].

Data quality procedures of ecosystem and floor GEM fluxes followed our previous study[3]. First, to avoid trap biases, 5-min GEM concentration measurements were first separated by sampling traps (note that Tekran analyzers have two gold sampling traps, which alternate collection and measurement of GEM in air). Then, GEM concentrations measured by each trap were averaged for each of the two vertical measurements heights to calculate 30-min average gradients, and finally gradients for both traps were averaged (Fig. S7). We performed outlier removal of the raw dataset of 30-min GEM concentration gradient and removed data when conditions were highly

stable or unstable (z/L < −2 or z/L > 1), followed by a second outlier removal procedure for calculated GEM fluxes. Missing data were interpolated using median hourly values of each respective month (supplementary dataset 1). To independently verify the flux-gradient approach for above-canopy measurements, $CO_2$ fluxes calculated by the flux-gradient approach were compared to fluxes measured by the EC method for 1 year, and good agreement was observed in magnitude and direction of $CO_2$ fluxes using the two approaches (Fig. S8). No independent flux verification was possible for forest floor fluxes, however.

Random error propagation of ecosystem-level GEM fluxes was performed using a daily-differencing approach described by Hollinger and Richardson[54]. Flux uncertainties were propagated when calculating cumulative fluxes by using the frequency distribution of random errors (Fig. S9), as described in more detail in Text S3.

## Vegetation, precipitation, throughfall, and snow sampling and analysis

At Howland Forest, vegetation samples were collected from the dominant tree species of red spruce, eastern hemlock, and red maple (including leaf/needle, root, bark, and bole wood), and moss. All samples (collected in triplicate) were dried in a stainless steel oven at 65 °C until constant weight, ground, and measured for Hg concentrations using a Direct Mercury Analyzer (DMA-80, Milestone). Precipitation, throughfall, and snow samples were manually collected several times during the period of flux measurement, including four times for wet-only and throughfall collections and four times for fresh snow and snowpack collection under the canopy and in the open field. Rain and melted snow samples were filtered through 0.45 µm filter membranes in the field and preserved by adding trace metal grade HCl (0.5%). Hg concentrations of these samples were measured by a Mercury Analysis System (Model 2600, Tekran Inc.) as detailed in a previous study[55]. Wet, throughfall, and litterfall deposition fluxes were estimated using Hg concentrations in samples multiplied by mass of rainfall and throughfall and mass of litterfall (10-year average).

## Data availability

All data included in this manuscript are available in Supplementary Information and in the associated data paper[3].

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

## Acknowledgements

This study was supported by the US National Science Foundation (AGS award no. 1848212 (D.O.) and DEB award no. 2027038 (D.O.)).

## Author contributions

J.Z. and D.O. contributed to the writing and overall conceptualization. J.Z. performed the observation and prepared the figures. J.Z., S.W.B., and E.M.R. analyze the data. D.Y.H. provided the meteorological data. D.Y.H. and T.W. contributed to the editing. J.T.L. helped sampling in the field. D.O. designed the study.

## Competing interests

The authors declare no competing interests.
