## [Peer Review File · Nature Communications]

Comparing ecosystem gaseous elemental mercury fluxes over a deciduous and coniferous forestReviewer #1 (Remarks to the Author):

The manuscript reports on a comparative long-term study of Hg uptake at a landscape level between a deciduous and a coniferous forest. The study is carried out with micrometeorological technology, which is crucial for being able to determine fluxes at the ecosystem level. However, the comparison is based on observational sets that are one behind the other in time. Using gradient-based micrometeorological methods over forest is controversial, as the measurements take place in the roughness sublayer. Alternative methods such as REA are perhaps better suited for this purpose. Readers should be informed of this. The study is one of very few studies of forest ecosystem air Hg exchange over a long period of time that has been reported. A literature search shows that a study of similar design has been carried out in southern China. Although the study leaves much more to be desired and is less representative (young forest affected by air pollution), perhaps it should be discussed. The same goes for a 2004-2005 study of hardwood forests in Connecticut. It should be pointed out that previous studies did not take exchange with the forest floor into consideration in a satisfactory way.

Main conclusions and claims are well supported.

The language flows well but sometimes the discussion lacks sufficient depth.

The measurements seem to have involved all the desirable QA/QC (flux-gradient vs. EC, line bias, error estimation, regular calibrations, etc.) and the footprint fetch seems unlimited. But the account is scant when it comes to data coverage and the processing of raw data. What corrections were made when constructing the 30-min data? How were periods of near-zero or negative eddy diffusivity treated?

Several of the references to Figures are incorrect (e.g. L58-59).

Furthermore, some of the figures contain typos and have colors that are not suitable for colorblind people.

Reviewer #2 (Remarks to the Author):

Review for "Comparing ecosystem gaseous elemental mercury fluxes over a deciduous and coniferous forest" by Zhou et al. (NCOMMS-22-41941-T)

The manuscript compares the air-surface exchange fluxes of mercury at two forest sites in the Northeast US. The two study sites are of different forest types (deciduous and coniferous). The measurements at the coniferous forest site are of new data sets; and the data at the deciduous forest site had been published previously for comparison. Although the manuscript does not provide specific details, the measurements and instrumentation at both sites are said to be similar. The GEM fluxes were measured using micro-metrological methods above the canopies. Additional measurement of mercury concentrations for biomass, precipitation and snow samples was also made. It is not clear the if air-soil exchange fluxes (and under what measurement conditions) were measured above the forest floors, although the floor GEM fluxes were reported (Fig. 3 and S6). The diurnal and seasonal variations of the measurement, as well as the cumulated fluxes of Hg and CO₂, are presented.

The manuscript provides comprehensive datasets of mercury fluxes of the two forest ecosystems, along with the CO₂ measurements at the ecosystem level, which is a nice addition to the existing datasets for the mercury research community. However, the manuscript falls short to present "important advances of significance" in the understanding of biogeochemical cycling of mercury in the forest ecosystem. It is unclear what the primary scientific objectives are, and the principal conclusions are derived from the work. Such a study is perhaps more relevant for journals that document scientific data for reference, not Nature Communications.

Specific comments for authors to consider revision for future submissions:

1. The manuscript can benefit from a clear set of scientific objectives in the introduction section. Mercury fluxes in forest ecosystem can be highly variable due to the weather conditions, site differences and measurement uncertainties. Based on the data presented in the manuscript (slightly more than a year at different time at two sites), it is not clear if the data are directly comparable, not to mention that the year-to-year variability at each site is not characterized. Given such a circumstance, it is not clear if the comparison is meaningful and conclusive. The primary conclusions of the manuscript are site specific, not scientific understandings that can be applied generally, and therefore somewhat lack innovations.

2. The SI can be more descriptive on how and what measurements were made, with specified data quality metrics and uncertainties. Micro-meteorological measurements of GEM fluxes are no easy tasks and biases can occur by small changes of experimental setups. Statistics of the measurements show also be shown.

3. It is not clear how the mass balance was constructed. Based on the data presented in Table 2, it is somewhat difficult to determine if the mass balance is closed. The graphic quality of Fig. 1 can be improved. It is unclear what messages Fig. 1E and 1F convey. Fig. 2 clearly show difference of the two forest types. However, the factors that lead to the observed difference are not discussed thoroughly. It is not clear if the differences shown in each subplot of Fig. 3 are significant.

Reviewer #3 (Remarks to the Author):

This study provides a valuable data set of GEM deposition flux over two forest sites because it is very challenging in collecting GEM flux data. In-depth analysis of the data is also presented such as interpreting GEM flux using CO₂ flux data.

I have a few (mostly minor) concerns for the authors to consider as explained below:

Lines 30-32: This sentence may not be very accurate. Throughfall should mainly washout particulate-bound mercury and probably a small portion of gaseous oxidized mercury collected on to the leaves and then deliver them to the underlying soil, and this process likely transfer little GEM to the underlying soil. Litterfall is indeed the main mechanism transferring GEM to the soil surface.

Lines 74-75: Regarding the differences in the annual GEM deposition between the two forests: can the authors provide any insight if any portion of the differences were caused by the different ambient GEM concentration and/or leaf area index? These two variables are the most important parameters in modeling GEM dry deposition so such information would be very useful for the scientific community.

Line 22-228: What are the proportions of highly stable and unstable micrometeorological conditions? Were the flux positive or negative under each of these conditions? The data selection criteria can have very big impact on the calculated annual flux. For example, if a larger portion of data having positive fluxes (emission) were removed than were those having negative fluxes, the estimated annual deposition would be biased higher. I asked this question because I found the GEM deposition presented here (especially for the deciduous forest at 25.1 $\mu\text{g m}^{-2}$) is extremely high compared to what have been reported in literature.

What are the relative magnitudes between the sensitivity of the instrument measuring GEM and the GEM gradient? In other words, how much confident do you have if the observed bi-directional exchange is actually caused by the real GEM gradient instead of by measurement uncertainties (especially in the case for quantifying soil flux inside the

canopy)?

Response to Reviewers' comments on "Comparing ecosystem gaseous elemental mercury fluxes over a deciduous and coniferous forest" (manuscript NCOMMS-22-41941-T)

We thank the three anonymous reviewers and the journal editor for their detailed feedback. We addressed all comments of the reviewers and editor in this revised manuscript. Because we significantly changed the structure of the manuscript in response to reviewer and editor comments, we did not track all detailed changes of the original manuscript. However, if the editor or reviewers would like to see tracked changes, we are happy to provide a version with all annotated changes.

Reviewer #1 Comments on manuscript NCOMMS-22-41941-T

Comment 1: The manuscript reports on a comparative long-term study of Hg uptake at a landscape level between a deciduous and a coniferous forest. The study is carried out with micrometeorological technology, which is crucial for being able to determine fluxes at the ecosystem level. However, the comparison is based on observational sets that are one behind the other in time. Using gradient-based micrometeorological methods over forest is controversial, as the measurements take place in the roughness sublayer. Alternative methods such as REA are perhaps better suited for this purpose. Readers should be informed of this.

Response: We now mention other micrometeorological methods to measure GEM fluxes, including eddy covariance (EC) and relaxed eddy accumulation (REA), and gradient-based methods such as the aerodynamic gradient (AGM) and modified Bowen ratio, MBR (lines 256-261). Additionally, we clarify that we conducted a detailed method verification using ecosystem-level (i.e., above canopy) CO₂ fluxes measured both by the flux-gradient method with corresponding eddy covariance measurements at both measurement sites. Both diurnal variation in growing seasons and cumulative annual CO₂ fluxes during the full time of measurements show high agreement between the two methods (Fig. S8 and Text S2).

Yes, it is correct that the observational sets are one behind the other in time, and not conducted simultaneously. Corresponding deployments at two forests would not have been possible with the enormous servicing, calibration, and QC/QA needs of these measurements. We clarified this in the discussion of site differences (i.e., that reasons for discrepancies include potential inter-

annual variability as measurements were not conducted simultaneously, line 121-133).

Comment 2: The study is one of very few studies of forest ecosystem air Hg exchange over a long period of time that has been reported. A literature search shows that a study of similar design has been carried out in southern China. Although the study leaves much more to be desired and is less representative (young forest affected by air pollution), perhaps it should be discussed. The same goes for a 2004-2005 study of hardwood forests in Connecticut. It should be pointed out that previous studies did not take exchange with the forest floor into consideration in a satisfactory way.

Response: Thanks for the reviewer's suggestion. We added the discussion of the previous study conducted in a young forest affected by air pollution in southern China and a rural hardwood forests in Connecticut in lines 45-48.

Comment 3: Main conclusions and claims are well supported. The language flows well but sometimes the discussion lacks sufficient depth.

Response: We expanded our discussion, also in response to the other reviewer comments, e.g., in lines 75-78, 95-98, 121-133, 194-201, and 204-219.

Comment 4: The measurements seem to have involved all the desirable QA/QC (flux-gradient vs. EC, line bias, error estimation, regular calibrations, etc.) and the footprint fetch seems unlimited. But the account is scant when it comes to data coverage and the processing of raw data. What corrections were made when constructing the 30-min data? How were periods of near-zero or negative eddy diffusivity treated?

Response: We clarified how we processed data and calculation of gradients and fluxes (e.g., first data was separated by sampling traps, then, GEM concentrations measured by each trap were averaged for each vertical measurements to calculate 30-minute average gradients, then gradients for both traps were averaged, lines 297-303). The QA/QC is detailed in lines 38-64 and Text S1. Finally, we clarify data coverage of the data (i.e., 71% data coverage of the total deployment period), and we provide further information on gap filling of missing data in lines 304-305.

Comment 5: Several of the references to Figures are incorrect (e.g. L58-59).

Response: We revised and corrected references to Figures in lines 75.

Comment 6: Furthermore, some of the figures contain typos and have colors that are not suitable for colorblind people.

Response: Thanks for pointing out this. We revised the typos and changed colors in all figures.

Reviewer #2 Comments on manuscript NCOMMS-22-41941-T

Comment 1: The manuscript compares the air-surface exchange fluxes of mercury at two forest sites in the Northeast US. The two study sites are of different forest types (deciduous and coniferous). The measurements at the coniferous forest site are of new data sets; and the data at the deciduous forest site had been published previously for comparison. Although the manuscript does not provide specific details, the measurements and instrumentation at both sites are said to be similar. The GEM fluxes were measured using micro-metrological methods above the canopies. Additional measurement of mercury concentrations for biomass, precipitation and snow samples was also made. It is not clear if air-soil exchange fluxes (and under what measurement conditions) were measured above the forest floors, although the floor GEM fluxes were reported (Fig. 3 and S6). The diurnal and seasonal variations of the measurement, as well as the cumulated fluxes of Hg and CO₂, are presented.

Response: We clarified that we conducted flux-gradient flux measurement above the forest floors at each site to estimate forest floor GEM exchanges (including soil and litter). This methodology is the same as was used over the forest canopy for ecosystem-level GEM fluxes. We further detailed the methods of flux measurement and experimental set-up in lines 278-282.

Comment 2: The manuscript provides comprehensive datasets of mercury fluxes of the two forest ecosystems, along with the CO₂ measurements at the ecosystem level, which is a nice addition to the existing datasets for the mercury research community. However, the manuscript falls short to present “important advances of significance” in the understanding of

biogeochemical cycling of mercury in the forest ecosystem. It is unclear what the primary scientific objectives are, and the principal conclusions are derived from the work. Such a study is perhaps more relevant for journals that document scientific data for reference, not Nature Communications.

Response: We clarified the main scientific objectives in lines 53-57, i.e., to investigate forest-level GEM exchanges, how they partition into canopy and forest floor contributions, in both a deciduous and coniferous rural forest in northeastern USA; assess underlying mechanisms and drivers of GEM exchange, using spatial and temporal partitioning to infer about stomatal uptake (e.g. during daytime in canopies) and non-stomatal contributions (e.g., nighttime and forest floor exchange).

We disagree about the lack of important advances of significance. This study is the first time to compare how patterns and magnitudes of GEM exchange differs among forest types, and we provide new insights into drivers of ecosystem GEM exchanges at the ecosystem level, in particular canopy and non-canopy/soil contributions (which notably differ among forests). Critically, we provide insights into non-stomatal versus stomatal uptake (physiologic control of GEM exchange) and importance of nighttime uptake. We finally compare how partitioned GEM fluxes compare to indirect methods (particularly litterfall and throughfall estimation) which are widely used in the absence of direct GEM flux measurements. We hope that after our major revision, the reviewer also sees that the study provides transformative progresses for the community of the importance of sub-component fluxes and help focus future efforts in constraining GEM deposition uncertainties across global forests and ecosystems.

Specific comments for authors to consider revision for future submissions:

Comment 3: The manuscript can benefit from a clear set of scientific objectives in the introduction section.

Response: We added the clear set of the scientific objectives in the last paragraph of Introduction section as clarified above.

Comment 4: Mercury fluxes in forest ecosystem can be highly variable due to the weather conditions, site differences and measurement uncertainties. Based on the data presented in the

manuscript (slightly more than a year at different time at two sites), it is not clear if the data are directly comparable, not to mention that the year-to-year variability at each site is not characterized. Given such a circumstance, it is not clear if the comparison is meaningful and conclusive. The primary conclusions of the manuscript are site specific, not scientific understandings that can be applied generally, and therefore somewhat lack innovations.

Response: Yes, we agree that there are uncertainties due to meteorological conditions and some observed differences between forests may be due to inter-annual variability (and the lack of corresponding measurements, see response to reviewer above). We now discuss these limitations in the discussion.

However, we do not believe that these comparisons are meaningless. Our detailed analysis show repeatable seasonal and diel patterns across these forests across multiple months (Figures 1, 2 and S4). Error propagation based on random error analysis and standard deviation of the uncertainty of time-repeated measurements provided high confidence in the reliability of the cumulative GEM flux measurements. The differences in GEM flux records among forests (e.g., much earlier growing-season GEM deposition in the coniferous forest, 87% higher GEM deposition in the deciduous forest) between the coniferous forest and deciduous forest are striking. Even more striking are the strong differences in flux partitioning and nighttime/daytime deposition. While some of these measurements will need to be confirmed across further sites, this study points towards highly different deposition mechanisms across forests.

Comment 5: The SI can be more descriptive on how and what measurements were made, with specified data quality metrics and uncertainties. Micro-meteorological measurements of GEM fluxes are no easy tasks and biases can occur by small changes of experimental setups. Statistics of the measurements also be shown.

Response: Thanks for the reviewer's suggestion. We added more description on how measurements were performed and about the process of calculation uncertainties in SI, including Text S1: Performance of flux-gradient method to measure GEM fluxes, Text S2: Verification of flux-gradient method for CO₂ in comparison to Eddy Covariance fluxes, and Text S3: Random error estimation and error propagation of GEM flux measurements. We agree

that the micro-meteorological measurements of GEM fluxes are challenging, but we correspondingly measured the CO₂ fluxes and verified fluxes by comparison of flux-gradient method with Eddy Covariance CO₂ measurements (Figure S1). We provide statistics of measured gradients, analysis of random noise, and cumulative spectral distribution analyses of measured 30-minute GEM gradients using autocorrelations as detailed in Text S3 and Figure S10.

Comment 6: It is not clear how the mass balance was constructed. Based on the data presented in Table 2, it is somewhat difficult to determine if the mass balance is closed. The graphic quality of Fig. 1 can be improved. It is unclear what messages Fig. 1E and 1F convey. Fig. 2 clearly show difference of the two forest types. However, the factors that lead to the observed difference are not discussed thoroughly. It is not clear if the differences shown in each subplot of Fig. 3 are significant.

Response: We revised the text to better present the mass balance estimate in the manuscript and underlying approach using data from litterfall, precipitation, and throughfall measurements and estimates of Hg(II), and PHg deposition (lines 194-201). We improved the quality of Fig. 1 and moved 3D-figure subpanel (Fig. 1E, F) to SI.

We compared the 30-min. fluxes by paired t-test between the two forests, which showed significant differences ($p < 0.01$) (lines 91-93). Additionally, we have added paired t-test comparisons to show when hourly flux data are statistically significant between the two forests (*) in the Figure S11, because Figure 3 is already busy and not well suited to provide statistical comparison between the two sites. However, in addition to these tests, we provide an in-depth discussion on interesting similarities and dissimilarities of diel patterns based on these figures in the text. Some differences we are pointing out are that: (i) diel patterns of forest floor patterns are inherently different, showing deposition at the deciduous forest and emissions at the coniferous forest; nighttime patterns (non-stomatal uptake) are different among sites, showing 4% at the coniferous forest but nearly 1/4 at the deciduous forest; and other flux components (particular canopy uptake) are surprisingly similar and consistent among sites showing similar diel patterns.

Reviewer #3 Comments on manuscript NCOMMS-22-41941-T

This study provides a valuable data set of GEM deposition flux over two forest sites because it is very challenging in collecting GEM flux data. In-depth analysis of the data is also presented such as interpreting GEM flux using CO₂ flux data.

Response: Thanks for the reviewer's constructive comments.

I have a few (mostly minor) concerns for the authors to consider as explained below:

Comment 1: Lines 30-32: This sentence may not be very accurate. Throughfall should mainly washout particulate-bound mercury and probably a small portion of gaseous oxidized mercury collected on to the leaves and then deliver them to the underlying soil, and this process likely transfer little GEM to the underlying soil. Litterfall is indeed the main mechanism transferring GEM to the soil surface.

Response: Thanks for your comment. We revised this and deleted the throughfall in the sentence.

Comment 2: Lines 74-75: Regarding the differences in the annual GEM deposition between the two forests: can the authors provide any insight if any portion of the differences were caused by the different ambient GEM concentration and/or leaf area index? These two variables are the most important parameters in modeling GEM dry deposition so such information would be berry useful for the scientific community.

Response: We have compared the GEM concentrations in the two forests, which are 1.10 ng m⁻³ in the deciduous forest and 1.03 ng m⁻³ in the coniferous forest, showing no significant differences across the measurement season (Fig. S6). Additionally, we added leaf area index (LAI) in Figure 2 and now discuss the significant correlation observed between LAI and GEM fluxes at both forests (lines 75-78).

Comment 3: Line 22-228: What are the proportions of highly stable and unstable micrometeorological conditions? Were the flux positive or negative under each of these conditions? The data selection criteria can have very big impact on the calculated annual flux. For example, if a larger portion of data having positive fluxes (emission) were removed than

were those having negative fluxes, the estimated annual deposition would be biased higher. I asked this question because I found the GEM deposition presented here (especially for the deciduous forest at $25.1 \mu\text{g m}^{-2}$) is extremely high compared to what have been reported in literature.

Response: We clarified our outlier removal process and gap filling. We performed an outlier removal of the raw dataset of 30-min GEM concentration gradients and removed data when conditions are highly stable or unstable ($z/L < -2$ or $z/L > 1$) as flux measurements can be invalid under these conditions. We then performed a second outlier removal of the calculated GEM flux dataset. We provide the data coverage of our data over the time of measurements (71% of coverage). Please note that missing data were interpolated using median hourly values for each respective month of measurements – using this method, we can avoid biases due to data removal amounts for different days of the year or hours of the day.

Comment 4: What are the relative magnitudes between the sensitivity of the instrument measuring GEM and the GEM gradient? In other words, how much confident do you have if the observed bi-directional exchange is actually caused by the real GEM gradient instead of by measurement uncertainties (especially in the case for quantifying soil flux inside the canopy)?

Response: This is an excellent point and we provide additional clarification on this point. We clarified (SI lines 31-32) that measurements of individual pairs of GEM gradients were indeed largely below the detection limit of the GEM analyzer (about 0.05 ng m^{-3} based on 3 standard deviations of 5-minute resolution measurements) and flux measurements (e.g., we estimate a high median GEM flux detection limit of $31 \text{ ng m}^{-2} \text{ hr}^{-1}$ at 30-resolution based three standard deviations of paired fluxes using a daily-differencing method (lines 33-37). So the reviewer is absolutely correct that the presence of large bi-directional exchanges as shown in high-resolution data (Figure 1A and 1B) is indeed caused by measurement uncertainties (see discussions in lines 121-133).

However, by having time-extended measurements, we can employ time-averaging of data, e.g., using monthly hourly means and medians (e.g., 60 flux data points) to reduce variability and delineate diel patterns and daily time using averaging for daily fluxes (e.g., 48 flux data points). We provide additional statistics that the variance of our measurements is distinctly and

statistically different of random noise and that show that cumulative spectral distribution analyses and autocorrelations show a structure of the data that is linked to diel flux patterns (and not random) (Figure S10).

Reviewer #2 (Remarks to the Author):

The authors did a commendable job in revising the manuscript to address the previous review concerns. The revised manuscript is much more readable and has improved presentation of the data. Although the data provided in the manuscript are of benefit to Hg research community, I respectfully disagree with the authors that the quantification of seasonal and diel patterns of Hg(0) flux over two forest sites represents a significant innovation. The initial finding of flux difference over various forest types was reported back in early 2000s. Since then, field data and laboratory measurements by the research groups in North America, Europe, and East Asia (by this author group, and the work led by Gustin, Feng, Bishop, Sommar, Osterwalder, to name a few) have reported flux intensity and seasonal characteristics vary over different canopy covers. I would be perhaps useful for the authors to point out the specific advancement of this work in relation to previous measurement efforts.

Overall, I recommend that the manuscript be accepted after minor revision and congratulate the authors on this contribution.

Reviewer #3 (Remarks to the Author):

The authors have addressed my comments I previously provided and I feel the manuscript have been improved and may be accepted for publication. After reading the revised manuscript and the authors' responses to other reviewers' comments, I now only have one additional suggestion for the authors to consider:

Line 194-201: NADP Hg wet deposition data is discussed in this paragraph. I feel discussing NADP Hg dry deposition, although from model estimates as provided in Zhang et al. (2016 EST 50, 12864-12873), might be more useful here. Besides, the model estimates from this reference may be useful in responding concerns provided by reviewer #2 (comment 4) sine the model estimates of Hg dry deposition cover multiple years and multiple forest sites. Note that the model estimates have been constrained to minimize uncertainties using four different approaches and are believed to be reliable.

Line 192 Correct "Bass balance" to "Mass balance"

Response to Reviewers' comments on "Comparing ecosystem gaseous elemental mercury fluxes over a deciduous and coniferous forest" (manuscript NCOMMS-22-41941-T)

We have incorporated reviewers' recommendations in the revised manuscript via point-by-point response. The corresponding changes relating to the reviewers' comments have been marked in blue fonts in the revised manuscript.

Reviewer #2 Comments on manuscript NCOMMS-22-41941A

Comment 1: The authors did a commendable job in revising the manuscript to address the previous review concerns. The revised manuscript is much more readable and has improved presentation of the data. Although the data provided in the manuscript are of benefit to Hg research community, I respectfully disagree with the authors that the quantification of seasonal and diel patterns of Hg(0) flux over two forest sites represents a significant innovation. The initial finding of flux difference over various forest types was reported back in early 2000s. Since then, field data and laboratory measurements by the research groups in North America, Europe, and East Asia (by this author group, and the work led by Gustin, Feng, Bishop, Sommar, Osterwalder, to name a few) have reported flux intensity and seasonal characteristics vary over different canopy covers. I would be perhaps useful for the authors to point out the specific advancement of this work in relation to previous measurement efforts. Overall, I recommend that the manuscript be accepted after minor revision and congratulate the authors on this contribution.

Response: We really appreciate the constructive comments that significantly improve the manuscript. Yes, we agree with the reviewer that many studies have investigated the mercury fluxes in the forests and contributed significantly to understand the role of global mercury cycling. We have added these references and then pointed out the specific advancement of our work in lines 39-42. I hope the revised manuscript addressed all the comments and meet the publishing standard.

Reviewer #3 Comments on manuscript NCOMMS-22-41941-T

The authors have addressed my comments I previously provided and I feel the manuscript have

been improved and may be accepted for publication. After reading the revised manuscript and the authors' responses to other reviewers' comments, I now only have one additional suggestion for the authors to consider:

Line 194-201: NADP Hg wet deposition data is discussed in this paragraph. I feel discussing NADP Hg dry deposition, although from model estimates as provided in Zhang et al. (2016 EST 50, 12864-12873), might be more useful here. Besides, the model estimates from this reference may be useful in responding concerns provided by reviewer #2 (comment 4) since the model estimates of Hg dry deposition cover multiple years and multiple forest sites. Note that the model estimates have been constrained to minimize uncertainties using four different approaches and are believed to be reliable.

Response: We really appreciate the reviewer's thoughtful suggestions in this round and last round. We have added the results and discussion of the dry deposition by Zhang et al. (2016 EST 50, 12864-12873) in lines 201-205.

Line 192 Correct "Bass balance" to "Mass balance"

Response: The typo was revised in line 193.